# Exploring Socio-Demographic Factors Affecting Psychological Symptoms in Humidifier Disinfectant Survivors

**DOI:** 10.3390/ijerph182211811

**Published:** 2021-11-11

**Authors:** Hye-Yun Ko, Seung-Hun Ryu, Min-Joo Lee, Hun-Ju Lee, Soo-Young Kwon, Seong-Mi Kim, Sang-Min Lee

**Affiliations:** 1Department of Education, Korea University, 145 Anam-ro, Seongbuk-gu, Seoul 02841, Korea; kohyeyeon@gmail.com (H.-Y.K.); mjbravo@korea.ac.kr (M.-J.L.); 2Humidifier Disinfectant Health Center, National Institute of Environmental Research, 42 Hwangyong-ro, Seogu, Incheon 22689, Korea; greenkorea247@korea.kr (S.-H.R.); seongmi@korea.kr (S.-M.K.); 3University Industry Foundation, Yonsei Humidifier Disinfectant Health Center, Yonsei University, 50, Yonsei-ro, Seodaemun-gu, Seoul 03722, Korea; hunjulee@yonsei.ac.kr; 4United Graduate School of Thelogy, Yonsei Humidifier Disinfectant Health Center, Yonsei University, 50, Yonsei-ro, Seodaemun-gu, Seoul 03722, Korea; sykwon@yonsei.ac.kr

**Keywords:** humidifier disinfectant survivors, psychological symptoms, socio-demographic factors, social disaster, public health

## Abstract

This study aimed to compare the psychological symptoms of humidifier disinfectant survivors to the general population and explore socio-demographic factors influencing survivors’ psychological symptoms. A one-way Multivariate Analysis of Covariance (MANCOVA) and a series of two-way MANCOVA were conducted with a sample of 228 humidifier disinfectant survivors and 228 controls. The results demonstrated that the survivor group displayed higher anxious/depressed symptoms, withdrawn symptoms, somatic complaints, thought problems, attention problems, aggressive behavior and rule-breaking behavior than the general group. Moreover, among the socio-demographic factors, the two-way interaction effects of group × family economic status and group × number of friends were found to be statistically significant. The limitations and implications of this study are discussed.

## 1. Introduction

In April 2011, media reported that mothers had a mysterious disease at Asan Medical Center in Seoul, South Korea [1]. Six women were hospitalized in the respiratory intensive care unit before and after childbirth. The symptoms were respiratory failure and pulmonary fibrosis [2]. The patients were not from one area, but from all over Korea. The Korea Center for Disease Control and Prevention (CDC) commissioned an epidemiological investigation to determine the cause of these unique symptoms. The results showed that the humidifier disinfectant, a biocide used by adding to the water in the humidifier, was causing these diseases [1,2]. According to the CDC, humidifier disinfectants are absorbed into the body through the nose, mouth and skin during respiration, which has a deleterious effect on lung disease [2].

Approximately 9.98 million humidifier disinfectant products were sold from 1994 to 2011; these were distributed to unspecified people [3]. The chemicals in the humidifier disinfectant were poly hexamethylene guanidine phosphate (PHMG), Oligo 2-(2-ethoxy) ethoxy ethyl guanidine chloride (PGH), cholromethylisothiazolinone (CMIT) and methylisothiazolinone (MIT), among others. Among them, products containing CMIT/MIT, the most toxic chemicals, have been sold since 1994 [4].

Diseases caused by toxic chemicals take a long time to show if the toxic chemicals have caused actual health damage, as they involve many variables, such as the incubation period [5]. It is difficult to accurately count the number of survivors of humidifier disinfectant damage. As of September 2021, the government reported that the number of persons who claimed damage was 7540, including 1713 deaths [6]. 

The humidifier disinfectants accumulate in the body, damage lung cells and ultimately lead to widespread health impairments. Pulmonary fibrosis, asthma and dyspnea are the main symptoms [7]. In addition, acute bronchitis, pneumonia, rhinitis, interstitial pneumonia, nonspecific otitis media, chronic sinusitis, chronic obstructive pulmonary disease, acute sinusitis and acute bronchiolitis have been reported [8]. In a study on mice, it is reported that lung damage caused by repeated exposure to PHMG-P does not recover from structural changes caused by pulmonary dysfunction and pathology, even after a long recovery period. This suggests that the damage caused by humidifier disinfectants may also have long-term effects on survivors. Therefore, plans to help survivors recover need to be considered in a long-term context [9].

In addition to physical health, the damage caused by humidifier disinfectant adversely affects mental health problems. As for the mental health problems that occurred after exposure, depression and helplessness were reported by 57.5% of individuals, guilt and self-blame by 55.1%, anxiety and tension by 54.3%, suicidal thoughts by 27.6% and suicide attempts by 11%. This indicates that survivors’ suicide attempts are 4.5 times higher than the general population [8]. The deterioration of physical health impacts the survivor, whereas mental health problems impact both the survivor and his/her significant others [10]. Moreover, studies have shown that socio-demographic, social and environmental factors should be considered in the psychological factors of victims of social disasters, such as social support, sociality and community cohesion [11,12,13]; this indicates that the psychological pain of survivors should be understood from a socio-demographic perspective. Humidifier disinfectants can be classified as a social disaster and it is closely related to Japan’s exposure to radiation in Fukushima, in that it had many random victims not only physically but also psychologically [14].

Studies argue that the mental and physical health of survivors are much more serious than those of the general group and require special attention [6,10]. It is necessary to explore more clearly which variables are specifically related to survivors’ psychological difficulties. This study aimed to examine the psychological health of humidifier disinfectant survivors and the general population. Next, we examined the socio-demographic variables influencing survivors’ psychological symptoms. Specifically, this study attempts to explore survivors’ psychological symptoms by gender, economic status, educational level and social variables such as the number of friends. This study can serve as preliminary research to identify important variables when designing intervention strategies to improve survivors’ mental health issues. 

## 2. Method

### 2.1. Participants

This study was approved by the National Institute of Environmental Research (NIER) in South Korea. The data of humidifier disinfectant survivors were collected through an online survey using the Adult Self Report (ASR). A total of 228 survivors who were suffering from humidifier disinfectant damage participated in this survey. The mean age of participants was 42.23 years, with 10.90 standard deviation; 83 (36.4%) were male and 145 (63.6%) were female. To compare the psychological symptoms between the humidifier disinfectant survivor group and the general group, the norm data of ASR were utilized. The HUNO Inc. (ASR Provider Company in South Korea) provided 1003 norm data of ASR. Among the respondents in the general population, a random sampling method was used to select 228 participants. The mean age of participants in the general group was 37.86 years with 9.71 standard deviation; 120 (52.6%) were male and 108 (47.4%) were female.

### 2.2. Dependent Variables

#### Psychological Symptoms

To measure the psychological symptoms, the ASR, the Adult version of the Achenbach System of Empirically Based Assessment (ASEBA) [15], was used in this study. Along with MMPI-2, ASR is one of the most used personality assessments to measure psychological symptoms in South Korea. The ASR was validated in Korean and the convergent validity, concurrent validity and discriminant validity of the Korean version were confirmed [16]. The psychological symptoms scale consists of eight syndrome subscales and is rated on a 3-point Likert-type scale (0: “not true”, 1: “sometimes”, 2: “often true”). The eight syndrome subscales are as follows: anxious/depressed (18 items); withdrawn (9 items); somatic complaints (12 items); thought problems (10 items); attention problems (15 items); aggressive behavior (15 items); rule-breaking behavior (14 items); and intrusive (6 items). The combination of anxious/depressed, withdrawn and somatic complaints is termed internalizing problems, and the combination of aggressive behavior, rule-breaking behavior and intrusive is termed externalizing problems. Anxious/depressed subscale measures the feelings of being emotionally depressed, overly worried and anxious, and the sample items are “I worry about my future” and “I cry a lot”. The withdrawn subscale evaluates withdrawal, passive attitude and showing no interest in surrounding people, “I don’t get along with other people” and “My social relations with the opposite sex are poor” are the sample items. The sample items of the somatic complaints subscale, which assesses various physical symptoms despite no clear medical cause, are “I feel dizzy or lightheaded” and “I feel tired without good reason”. The thought problems subscale estimates unrealistic and bizarre thoughts and behaviors, such as excessive repetition of certain actions and thoughts and seeing phenomena or hearing sounds that do not exist. The sample items are “I can’t get my mind off certain thoughts” and “I hear sounds or voices that other people think aren’t there”. The attention problems subscale measures inattentive or hyperactive behavior and difficulty in making plans; the sample items are “I have trouble concentrating or paying attention for long” and “I daydream a lot”. The aggressive behavior subscale evaluates verbally or physically destructive behavior and hostile attitudes, with “I argue a lot” and “I blame others for my problems” being examples of the items in the subscale. The rule-breaking behavior subscale assesses impulsive engagement in problematic behaviors that do not follow rules or violate social norms at work or in society, and “I damage or destroy my things” and “I break rules at work or elsewhere” are the sample items. Sample items of the intrusive subscale that estimate behavior that bothers or disturbs others are “I brag” and “I try to get a lot of attention”. The internal consistency (Cronbach’s α) of the subscales ranged from 0.70 to 0.92 in this study. 

### 2.3. Independent Variables

Four variables were utilized as independent variables: gender, educational level, family economic status and number of friends. Gender was coded into two categories (male and female). Educational level was measured by one item asking about educational level, and was coded into three categories (graduate school graduate or higher than graduate school graduate, university graduate, high school graduate or less than high school graduate). Family economic status was assessed by one item, “compared to the economic level of all households in Korea, which of the following would you say you belong to?” This single question was coded into three categories (lower, middle and upper economic status). Number of friends was measured by one item, “how many friends do you have besides family?” This single question was coded into two categories (few: 0–3 friends; and many: 4 or more friends).

### 2.4. Covariate Variable

Because age is a continuous variable, we included age as a covariate. According to a previous study, age is related to psychological symptoms; younger adults (ages 18–35) scored significantly higher than older adults (ages 36–59) on anxious/depressed, somatic complaints, attention problems, aggressive behavior and intrusive [17].

### 2.5. Data Analysis

One-way Multivariate Analysis of Covariance (MANCOVA) was conducted to compare dependent variables between the general and survivor groups. Moreover, Cohen’s *d* was calculated to determine the effect size for differences between the general and survivor groups. Effect sizes are considered small if *d* = 0.2, medium if *d* = 0.5 and large if *d* = 0.8 [18]. Then, a series of two-way MANCOVA was conducted to determine the main and interaction effects of independent variables on dependent variables. The independent variables were gender, educational level, family economic status and number of friends. The dependent variables were anxious/depressed, withdrawn, somatic complaints, thought problems, attention problems, aggressive behavior, rule-breaking behavior and intrusive. The covariate variable was age. As eight dependent variables were conceptually related to each other (average *r* = 0.57), the MANCOVA, which controlled correlations among dependent variables, was suitable for the analysis.

## 3. Results

A one-way MANCOVA with group as the independent variable, age as the covariate and psychological symptoms as dependent variables was performed to compare the psychological symptoms of the general and survivor groups. Moreover, Cohen’s *d* was calculated to determine the effect size for differences between the two groups. The covariate of age (Wilks’ lambda = 0.875; F(8,446) = 7.967; *p* < 0.001; η^2^ = 0.125) was statistically significant. Moreover, a significant main effect for the group (Wilks’ lambda = 0.661; F(8,446) = 28.574; *p* < 0.001; η^2^ = 0.339) was found. Main effects were found for seven psychological symptoms: anxious/depressed, F(1,453) = 152.301, *p* < 0.001; withdrawn, F(1,453) = 143.436, *p* < 0.001; somatic complaints, F(1,453) = 120.920, *p* < 0.001; thought problems, F(1,453) = 80.489, *p* < 0.001; attention problems, F(1,453) = 56.567, *p* < 0.001; aggressive behavior, F(1,453) = 86.133, *p* < 0.001; and rule-breaking behavior, F(1,453) = 9.383, *p* < 0.01. The results revealed that the survivor group displayed higher anxious/depressed symptoms, withdrawn symptoms, somatic complaints, thought problems, attention problems, aggressive behavior and rule-breaking behavior than the general group (see Table 1). Group differences in anxious/depressed, withdrawn, somatic complaints, thought problems and aggressive behavior indicated a large effect size (Cohen’s *d* > 0.80).

Next, a series of two-way MANCOVA were conducted separately, with age as the covariate. First, the two-way interaction effect of group × gender was tested. The covariate of age (Wilks’ lambda = 0.874; F(8,444) = 8.005; *p* < 0.001; η^2^ = 0.126) was statistically significant. Moreover, a significant main effect was found for group (Wilks’ lambda = 0.681; F(8,444) = 26.002; *p* < 0.001; η^2^ = 0.319) and gender (Wilks’ lambda = 0.938; F(8,444) = 3.641; *p* < 0.001; η^2^ = 0.062). Specifically, main effects for gender were observed for somatic complaints, F(1,451) = 5.699, *p* < 0.05; and rule-breaking behavior, F(1,451) = 4.451, *p* < 0.05. Female participants displayed more somatic complaints than male participants. On the other hand, male participants showed more rule-breaking behavior than female participants. However, the two-way interaction effect of group × gender was statistically insignificant (Wilks’ lambda = 0.977; F(8,444) = 1.309; *p* = 0.237; η^2^ = 0.023). Second, the two-way interaction effect of group × educational level was tested. The covariate of age (Wilks’ lambda = 0.882, F(8,441) = 7.399; *p* < 0.001; η^2^ = 0.118) was statistically significant. A significant main effect for group (Wilks’ lambda = 0.760; F(8,441) = 17.427; *p* < 0.001; η^2^ = 0.240) was found. However, the main effect for educational level (Wilks’ lambda = 0.964; F(16,882) = 1.033; *p* = 0.418; η^2^ = 0.018) and the two-way interaction effect of group × educational level (Wilks’ lambda = 0.944; F(16,882) = 1.615; *p* = 0.059; η^2^ = 0.028) were statistically insignificant.

Third, the two-way interaction effect of group × family economic status was tested. The covariate of age (Wilks’ lambda = 0.876; *F*(8,442) = 7.837; *p* < 0.001; η^2^ = 0.124) was statistically significant. A significant main effect of group (Wilks’ lambda = 0.659; F(8,442) = 28.546; *p* < 0.001; η^2^ = 0.341) and family economic status (Wilks’ lambda = 0.883, F(16,884) = 3.559; *p* < 0.001; η^2^ = 0.061) was observed. Moreover, a significant two-way interaction effect of group × family economic status was found (Wilks’ lambda = 0.922; F(16,884) = 2.304; *p* < 0.01; η^2^ = 0.040). Specifically, interaction effects were observed for five psychological symptoms: anxious/depressed, F(2,449) = 4.615, *p* < 0.05; withdrawn, F(2,449) = 10.407, *p* < 0.001; thought problems, F(2,449) = 7.869, *p* < 0.001; attention problems, F(2,449) = 7.740, *p* < 0.001; and rule-breaking behavior, F(2,449) = 5.003, *p* < 0.01. Although the general group’s psychological symptoms decreased as the family economic status increased, the survivor group’s psychological symptoms showed a different tendency. In other words, the upper family economic status group had the highest mean value for psychological symptoms, followed by the lower and the middle groups (see Table 2 and Figure 1).

Fourth, the two-way interaction effect of group × number of friends was tested. The covariate of age (Wilks’ lambda = 0.868; F(8,444) = 8.428; *p* < 0.001; η^2^ = 0.132) was statistically significant. A significant main effect for group (Wilks’ lambda = 0.678; F(8,444) = 26.403; *p* < 0.001; η^2^ = 0.322) and number of friends (Wilks’ lambda = 0.860; F(8,444) = 9.049; *p* < 0.001; η^2^ = 0.140) was observed. In addition, a significant two-way interaction effect of group × number of friends was found (Wilks’ lambda = 0.961; F(8,444) = 2.243; *p* < 0.05; η^2^ = 0.039). The effects of two-way interaction were found on six psychological symptoms: anxious/depressed, F(1,451) = 7.069, *p* < 0.01; withdrawn, F(1,451) = 14.408, *p* < 0.001; somatic complaints, F(1,451) = 4.785, *p* < 0.05; thought problems, F(1,451) = 5.326, *p* < 0.05; attention problems, F(1,451) = 6.389, *p* < 0.05; and aggressive behavior, F(1,451) = 9.527, *p* < 0.01. Compared to the general group, which showed a slight decrease in psychological symptoms with an increase in the number of friends, the survivor group showed a prominent decrease in psychological symptoms as the number of friends increased (see Table 3 and Figure 2).

## 4. Discussion

The first purpose of this study was to compare the psychological symptoms of humidifier disinfectant survivors and general groups. To examine these differences, a one-way MANCOVA was performed. The survivor group showed higher scores on seven psychological symptoms than the general group: anxious/depressed, withdrawn, somatic complaints, thought problems, attention problems, aggressive behavior and rule-breaking behavior. Consistent with previous studies [19] that suggested that social disasters have a serious impact on the mental health of survivors, the mean differences in scores between humidifier disinfectant disaster survivors and general groups in this study indicated that survivors experienced severe psychological difficulties. It was also revealed that the greater the damage and the closer the relationship with the survivor, the more severe the psychological trauma of the survivors.

As for the social aspect of survivors, the findings of a meta-analysis study [20] showed that multilevel social support from the micro-system (i.e., family and friends), meso-system (i.e., neighborhood, community) and macro-system (i.e., society and culture) need to be implemented to help survivors’ recovery. Therefore, this study explored the effects of socio-demographic variables, such as gender, educational level, family economic status and number of friends on psychological symptoms. A series of two-way MANCOVAs was conducted to determine the main and interaction effects of four demographic variables by groups (survivors vs. general groups) on psychological symptoms. Among the four demographic variables, the results indicated that educational level had no main effect on participants’ psychological symptoms. Studies have found mixed results regarding educational levels and psychological symptoms. Some studies reported that low educational level was related to increased behavioral problems [21], but other studies found that educational level was not related to behavioral problems [22]. The findings of this study are consistent with the findings of latter studies; that is, the educational level of both humidifier survivors and the general group did not influence psychological symptoms.

We also examined the relationship between psychological symptoms, family economic status and number of friends as factors of social resources. Family economic status and number of friends had significant main and interaction effects on psychological symptoms. Interestingly, the interaction effects of family economic status and the two groups (i.e., survivors and general groups) were also found for five psychological symptoms: being anxious/depressed, withdrawn, thought problems, attention problems and rule-breaking behavior.

The psychological symptoms of the survivor group that were divided according to family economic status level showed a V-shaped pattern, while those of the general group showed a decreasing pattern as the family economic status level increased. Consistent with many prior studies that lower levels of family economic status was associated with higher levels of psychological symptoms [23,24], our findings also show that individuals with lower family economic status have more severe psychological symptoms. Next, survivors with high economic status have more severe psychological symptoms than those in other groups (i.e., middle and lower family economic status). The results of the higher family economic status group imply that social comparison may contribute to psychological symptoms. The results of the higher family economic status group could be explained by the theory of the ‘big fish little pond’ effect (BFLPE). Herbert and John emphasized the importance of the frame of reference with the BFLPE model. According to the model, individuals compare their own self-concept with their peers, and individuals have higher self-concept when they are in a less capable group than in a more capable group, even though they perform equally [25]. Although the BFLPE model was originally intended to explain academic achievement, it can be expanded and used to explain the psychological status experienced subjectively by individuals who compare relative satisfaction with those around them. Because survivors with a high economic status have more severe psychological symptoms than other groups, it is important that psychologists and other such professionals focus on individuals’ subjective psychological damage rather than objectively measured ones. The results provide implications for considering the subjective frame of reference for survivors by designing differential psychological interventions.

Next, there were significant main and interaction effects of the number of friends and two groups (i.e., survivors and general groups) on six psychological symptoms: anxious/depressed, withdrawn, somatic complaints, thought problems, attention problems and aggressive behavior. Our results found that the level of psychological symptoms in both groups reporting that the number of friends being four or more was significantly healthier than the group with three or fewer friends. The effects of the number of friends on psychological symptoms were more prominent in survivors than in the general groups. That is, friends’ support was more helpful for the psychological health of survivors than general groups. Consistent with previous studies [26], social relationships contributed to the psychological symptoms of survivors. According to a meta-analysis study examining factors influencing post-traumatic stress response after a disaster, social relationships were found to be beneficial in post-traumatic stress response in social disasters, but not in natural disasters. Studies [27,28] have reported that disaster survivors can improve their quality of life, and return to their daily lives if they receive social support. Based on the research results, it can be seen that formal and informal social support should be provided for the recovery of disaster survivors.

This study had some limitations. First, the sample of humidifier disinfectant survivors was relatively small (*n* = 228). Although the sample was a representative group of humidifier disinfectant survivors, the results should be cautiously interpreted; therefore, future studies should use larger samples. Second, due to the small number of survivors, analysis of various dimensions of variables such as dividing the number of friends, more specifically into three groups, was limited. Future studies are needed to diversify dimension of variables. Third, the current findings are limited to data from one self-reported psychological assessment. Previous studies have shown that a survivor’s quality of life is affected by factors such as demographic characteristics, physical health, psychological characteristics and social support in an integrated way. To expand the knowledge of survivors’ psychological status, it is necessary to analyze their relationship with psychological health and additional data, such as physical status, diagnosed disease, amount of damage compensation and degree of damage to the family member. Fourth, the results are obtained from cross-sectional data, and it is necessary to analyze the psychological health of survivors longitudinally in the future.

Despite these limitations, the results of this study highlight survivors’ ability to recover. The findings of this study are expected to provide information on psychological symptoms and aid in the provision of counseling for better outcomes in survivors of humidifier disinfectant disasters. Additionally, to recover and improve survivors’ quality of life, it is believed that a comprehensive support system is needed in consideration of psychological health, as well as economic and environmental aspects.

## Figures and Tables

**Figure 1 ijerph-18-11811-f001:**
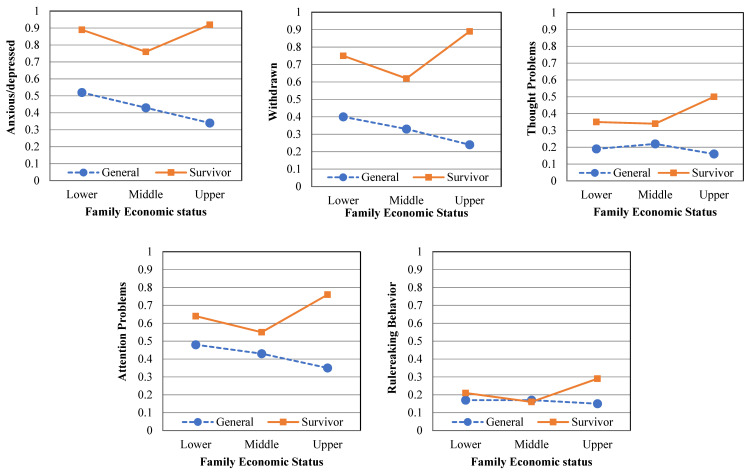
Mean scores for psychological symptom scores: according to family economic status.

**Figure 2 ijerph-18-11811-f002:**
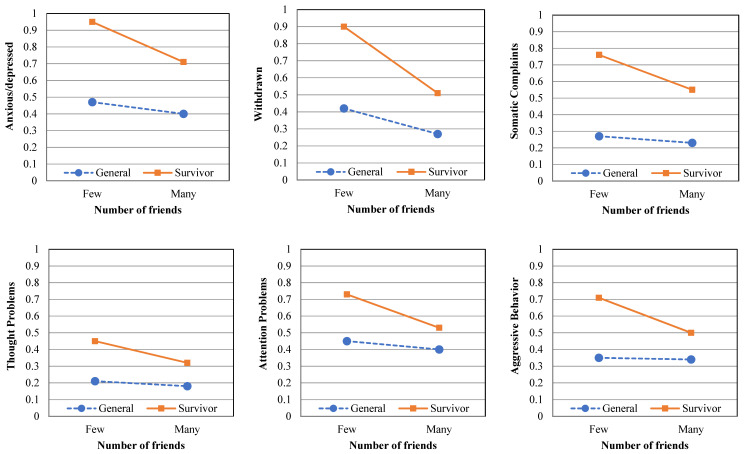
Mean scores for psychological symptom scores: according to number of friends.

**Table 1 ijerph-18-11811-t001:** Means (standard deviation) and Cohen’s *d* for psychological symptoms.

Dependent Variables	General Group (*n* = 228) Mean (SD)	Survivor Group (*n* = 228) Mean (SD)	Cohen’s *d*
Anxious/Depressed	0.43 (0.31)	0.85 (0.43)	1.12
Withdrawn	0.32 (0.33)	0.74 (0.42)	1.11
Somatic Complaints	0.25 (0.27)	0.67 (0.49)	1.06
Thought Problems	0.19 (0.18)	0.39 (0.30)	0.81
Attention Problems	0.42 (0.30)	0.64 (0.38)	0.64
Aggressive Behavior	0.34 (0.31)	0.62 (0.37)	0.82
Rule-Breaking Behavior	0.16 (0.19)	0.22 (0.24)	0.28

**Table 2 ijerph-18-11811-t002:** Means (standard deviation) for psychological symptom variables by group and family economic status.

Dependent Variables	General Group (*n* = 228) Mean (SD)			Survivor Group (*n* = 228) Mean (SD)		
	Lower (61)	Middle (97)	Upper (70)	Lower (61)	Middle (93)	Upper (74)
Anxious/Depressed	0.52 (0.33)	0.43 (0.29)	0.34 (0.32)	0.89 (0.44)	0.76 (0.39)	0.92 (0.46)
Withdrawn	0.40 (0.37)	0.33 (0.29)	0.24 (0.32)	0.75 (0.41)	0.62 (0.35)	0.89 (0.46)
Thought Problems	0.19 (0.16)	0.22 (0.20)	0.16 (0.18)	0.35 (0.30)	0.34 (0.22)	0.50 (0.36)
Attention Problems	0.48 (0.27)	0.43 (0.29)	0.35 (0.33)	0.64 (0.38)	0.55 (0.31)	0.76 (0.42)
Rule-Breaking Behavior	0.17 (0.17)	0.17 (0.19)	0.15 (0.22)	0.21 (0.24)	0.16 (0.18)	0.29 (0.29)

**Table 3 ijerph-18-11811-t003:** Means (standard deviation) for psychological symptom variables by group and number of friends.

Dependent Variables	General Group (*n* = 228) Mean (SD)		Survivor Group (*n* = 228) Mean (SD)	
	Few (83)	Many (145)	Few (133)	Many (95)
Anxious/Depressed	0.47 (0.34)	0.40 (0.30)	0.95 (0.43)	0.71 (0.39)
Withdrawn	0.42 (0.39)	0.27 (0.27)	0.90 (0.40)	0.51 (0.33)
Somatic Complaints	0.27 (0.31)	0.23 (0.24)	0.76 (0.52)	0.55 (0.41)
Thought Problems	0.21 (0.23)	0.18 (0.16)	0.45 (0.34)	0.32 (0.22)
Attention Problems	0.45 (0.32)	0.40 (0.28)	0.73 (0.39)	0.53 (0.34)
Aggressive Behavior	0.35 (0.30)	0.34 (0.32)	0.71 (0.39)	0.50 (0.32)

## Data Availability

The data presented in this study are available on request from the corresponding author.

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
