# Peer review of "Exploring Socio-Demographic Factors Affecting Psychological Symptoms in Humidifier Disinfectant Survivors"

_ijerph, 2021, doi:10.3390/ijerph182211811_

Round 1
Reviewer 1 Report
I think this is a very interesting study of the effects of a disaster on the psychological functioning of survivors. It isn't surprising that survivors exhibit psychological symptoms or that the number of friends may attenuate the relationship. What is surprising is that the highest economic level exhibits more symptoms among survivors. One thing that would be interesting would be to see whether the same relationships are found for an outcome variable that takes into account the common factor that is present in the dependent variables. So, the authors could extract a common factor by using factor analysis and see whether the same relationships are found, as the correlation between the different dependent variables is quite high. Another thing that would be interesting, and perhaps would be a more parsimonious approach, would be to run a hierarchical regression that would use the independent variables as predictors. So, rather than running a series of MANCOVAS, the authors could first enter whether the person is a survivor at the first level, then number of friends, age, gender, socioeconomic status at the second level, and then the second-order interactions between the independent variables at the third level. I believe that there would be a total of 15 independent variables, which given 456 participants (228 in each group), there would be an adequate number of observations for each independent variable (around 30 observations per predictor). I think this would be a better approach to analysing the data than the one that the authors have chosen.
Reviewer 2 Report
Dear Authors, thank you for the opportunity to read your article. I find the article well written and the presented research properly designed and conducted.
I leave a few points below for the Authors to consider.
- 1. Introduction - do the authors know about similar events and can they present their psychological consequences?
- 2.2. Dependent Variables - This part is sufficiently described. Sample Items an reliability of scales are presented.
- Do the Authors consider a different division of the number of friends: few, medium, many (similar to Family Economic Status)?
- Did the Authors find any gender differences in the dependent variable?
- Perhaps the authors could refer to any of these articles in the discussion: https://www.mdpi.com/search?q=Humidifier+Disinfectant
Good luck with your further research!
Round 2
Reviewer 1 Report
The authors have addressed my concerns, however, the quality of the English that is used needs to be improved.
Author Response
Professional editor in Editage company proofreads our manuscript. Please see the attachment.
